psychology/behaviour/health and disease and epidemiology

COVID-19, experiment, compliance, anti-contagion measures, Lombardy, Italy

**Author for correspondence:**
Mike Farjam
e-mail: mike.farjam@eu.lu.se

# Dangerous liaisons: an online experiment on the role of scientific experts and politicians in ensuring public support for anti-COVID measures

Mike Farjam[1], Federico Bianchi[2], Flaminio Squazzoni[2] and Giangiacomo Bravo[3,4]

[1]European Studies, Centre for Languages and Literature, Lund University, Lund, Sweden
[2]Department of Social and Political Sciences, University of Milan, Milan, Italy
[3]Department of Social Studies, and [4]Centre for Data Intensive Sciences and Applications, Linnaeus University, Växjö, Sweden

MF, 0000-0002-0882-4851; FB, 0000-0002-7473-1928

The effectiveness of public health measures to prevent COVID-19 contagion has required less vulnerable citizens to pay an individual cost in terms of personal liberty infringement to protect more vulnerable groups. However, the close relationship between scientific experts and politicians in providing information on COVID-19 measures makes it difficult to understand which communication source was more effective in increasing pro-social behaviour. Here, we present an online experiment performed in May 2020, during the first wave of the pandemic on 1131 adult residents in Lombardy, Italy, one of the world's hardest hit regions. Results showed that when scientific experts recommended anti-contagion measures, participants were more sensitive to pro-social motivations, unlike whenever these measures were recommended by politicians and scientific experts together. Our findings suggest the importance of trusted sources in public communication during a pandemic.

## 1. Introduction

During the first wave of the COVID-19 pandemic, the Lombardy region in Northern Italy was one of the hardest hit areas worldwide. Home to about 10 million people and a sixth of the total Italian population, the region had dramatic peaks of both infections and deaths cases during the first wave of the

pandemic, between February and May 2020, i.e. almost half of the 34 000 deaths in the whole country [1]. During the outbreak, the Italian government applied restrictive measures to contain the pandemic in Lombardy and in the whole country, including full isolation of entire areas, the closure of schools, public offices and most commercial activities. One of the first government initiatives was to set up a task force of public health scientific experts whose mission was to give advice on public policies. Scientific experts were often featured in the public media to explain anti-pandemic measures, while public decision-makers used the opinion of scientific experts to legitimate the imposed measures [2].

The Insights Unit of the WHO Regional Office for Europe [3] has suggested that measures infringing individual liberty and changing public habits to face a health emergency need to be perceived by all citizens as 'consistent, competent, fair, objective, empathetic or sincere', and must be 'communicated through trusted people and accessible channels'. Previous research on public health emergencies suggests that worst-case scenarios arise when measures requiring people's diffuse cooperation are announced and attempted without having strategies to enforce them, which involve efficient communication, public credibility of public decision-makers and attention to trust formation [4,5].

During the first wave of the pandemic, police records on regulation violations and data on regional mobility were occasionally reported in the media to indicate a substantial level of compliance by the Italian public [6,7], which was later confirmed by empirical research [8,9]. Note that this rate of compliance was initially not expected given weak enforcement and control by public institutions. Indeed, millions of citizens were required to lock-down at home, stop working, avoid social contacts and adopt hygiene-related measures, including most elements of the population with zero or minimal risk of suffering serious harm from the virus. Therefore, the cost of compliance with these measures was unequally distributed among different generations and social groups, in terms of psychological and social quality of life and availability of family support. For the most part, professionals, small entrepreneurs and shop retailers also suffered in terms of income reduction [10,11]. On the other hand, given that liberal democracies cannot enforce norms with invasive monitoring and social control on the whole population, free-riding could be a serious possibility and so a risk to mitigate [12]. Indeed, even if only a minority of the population did not follow anti-contagion measures, either publicly or at home, this could compromise efforts by everyone to protect vulnerable subjects and avoid hospitals' saturation [13] (a goal that unfortunately was only partially successful in Lombardy during the first wave of the pandemic [14]).

However, during a pandemic, as suggested in [15], the possibility of overcoming such cooperation traps that are intrinsic to widespread regulation compliance depends on pre-existing trust in experts, the capacity of governments to coordinate policy instruments and make choices about the degree of coercion, and the pro-social motivations of the public. As suggested by research on previous public health emergencies, public communication is key to develop trust and stimulate pro-social behaviour [16–18]. Indeed, in Italy, political leaders co-opted scientific experts during public announcements to promote compliance, as an incessant flow of news and expert comments on COVID-19 dominated all media. This was not limited to the most dramatic 'Phase 1'. The task forces of national and regional scientific experts were at the foreground also when determining plans for the gradual easing of measures during 'Phase 2', from 4 May to 14 June 2020. At the same time, recurrent announcements of the incoming 'Phase 2' during April 2020 was key to fulfil public expectations after weeks of sacrifice [19], scientific experts were constantly called on by politicians to reassure the public against uncertainty and (lack of) control.

Research suggests that how scientific evidence is framed can influence public perception [20], with potential negative consequences when public debate is *politicized* [21,22]. This occurs whenever the inherent uncertainty of scientific evidence is emphasized or omitted to promote a political agenda [23]. Using either a *political* or a *scientific* frame could therefore influence pro-active behaviour and policy support by the public [24,25]. Negative effects of science politicization on public trust have been found in research on health [26] and other 'hot' issues, such as climate change [21,27,28]. Thus, possibly due to the politicization and polarization of the public debate [29,30], it is not surprising that behavioural responses to calls for public compliance could well be sensitive to information signals and the source of communication [31,32].

However, the confusion of responsibilities and roles between scientific experts and politicians makes it difficult to assess who has the strongest communication effect on public behaviour [33,34]. Understanding whether politicians, scientists or a mix could be instrumental in promoting compliance with regulations is key to understanding how to manage the current waves of the pandemic [35]. Note that this would also apply to the current vaccination campaign that has been recently launched, when a large-scale cooperation will be needed to ensure maximum coverage.

To explore this, we considered the COVID-19 pandemic as the ideal context to perform an experiment on the role of scientific experts and public decision-makers as information signals of policy measures *in the making* [36]. Rather than using pre-existing research on social capital, trust and cooperative attitudes to estimate possible responses or designing a survey to measure public perceptions [17,37], we designed an online experiment on a sample of adult residents in Lombardy and launched it as the country was shifting towards partial easing of its current measures after months of lock-down. This was a key transition because: (i) there was uncertainty about the effect of the proposed changes; (ii) an exacerbation of measures in the case that the number of infections increased was still possible; and (iii) one of the key measures of 'Phase 2' included promoting a controversial smartphone application to trace people's contacts, provoking heated debate on privacy violation. Thus, we were specifically interested in understanding which source of information could promote pro-social attitudes and behaviour by the public. To do so, we measured participants' agreements on a set of enforcement measures but also actual pro-social attitudes and behaviour, including looking for extra information on these measures and voluntarily paying the cost for donating to COVID-19-related charities.

# 2. Methods

In our experiment, we manipulated the source of information when showing subjects a set of more restrictive measures related to the upcoming 'Phase 2'. These manipulations were minimal and only included differences in the source of background information in a vignette. Considering political distrust [38,39], the unpredictable effect of pre-existing performance of national and regional institutions on public compliance [40] and the front-line role played by scientific experts during the pandemic [41], we hypothesized that whenever measures were legitimized by scientific experts, subjects would be more inclined to support them, more interested to have extra information on them, and even more sensitive to the public benefit by donating to COVID-19-related charities.

## 2.1. Structure of the experiment

The experiment was implemented using oTree [42], which randomly assigned participants to the treatments. Starting with the treatment manipulation, participants were shown one of four vignettes, giving information on suggested measures to contain the COVID-19 pandemic during 'Phase 2'. The only difference between the four vignettes was the source of information regarding the recommended measures, i.e. politicians, scientific experts or both. These differences were minimized in that groups differed only in whether statements in the vignette were introduced such as 'According to [SOURCE]' or not. The source could either be (i) *scientific experts*, (ii) *politicians*, (iii) *scientific experts and politicians*, or (iv) *not specified*, the latter used as the baseline treatment. Immediately after being shown the vignette, participants were asked whether they wanted additional information regarding the statements in the vignette, which they could download at the end of the experiment. This was to measure the effect of manipulating the source of information on information-seeking behaviour.

In the second block of the experiment, participants were confronted with seven statements regarding suggested measures to contain the COVID-19 pandemic and were asked to rate them on an agreement–disagreement scale. These measures were those debated in the public media as possible counter-measures in the case of norm violation during 'Phase 2' and for social control, including the use of a contact-tracing smartphone app (see appendix A for a full list of these statements).

Furthermore, participants were informed that, as part of the experiment, they would receive a lottery ticket giving them a 1-in-50 chance to win a €50 Amazon voucher. Participants could then choose to instead donate the €1 value of the lottery ticket to a charity linked to COVID-19 (they were informed that this donation would be done by the researchers on their behalf). To avoid path-dependency between the order in which statements and the donation decisions were presented, the order of statement and donation tasks was randomized.

In the final block, participants were required to provide demographic information, were debriefed on the purpose of the experiment, were required not to discuss the content of the experiment for one week, and, finally, the lottery was resolved whenever participants chose to keep the lottery ticket.

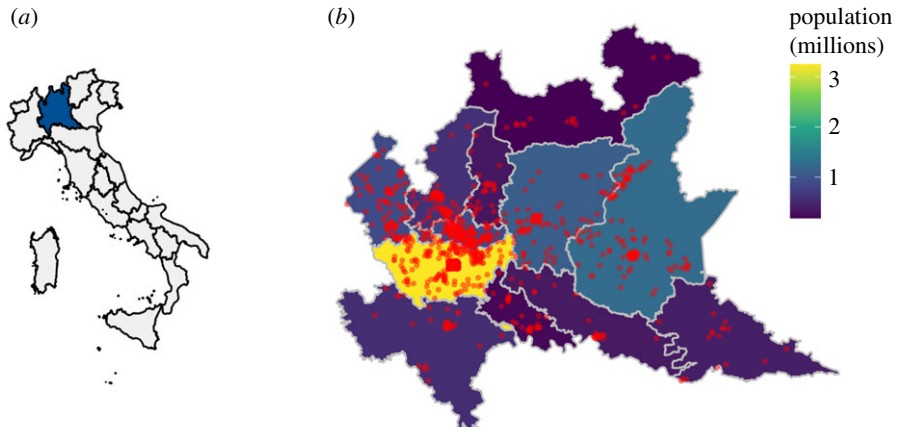

**Figure 1.** Geographical distribution of participants. Panel (*a*) shows Lombardy's location within Italy, while panel (*b*) includes the region's map. Dots indicate the approximate location of participants based on their self-declared municipality of residence. Background colours reflect the population of each province.

## 2.2. Measurements

Our analysis was focused on the three main decisions that participants had to make: (i) whether they donated their lottery ticket to support a COVID-19-related charity or not; (ii) whether they requested additional information on COVID-19 measures in the vignette or not; and (iii) to what extent they supported potential new COVID-19 public measures. As regards (i), participants had two options of COVID-19-related charities to support: Protezione Civile, a governmental organization which deals with the prediction, prevention and management of public emergencies, or ASST Fatebenefratelli Sacco, a regional governmental healthcare institution, which includes four hospitals and is a recognized research and teaching centre of excellence in many areas from infectious diseases to biomedical research in Lombardy.

We assumed that the participants' decision to give up the lottery in order to support these front-line organizations could be considered as a sign of pro-social behaviour. For (iii), the support of stricter measures to enforce compliance was measured through seven separate statements where participants had to indicate on a scale from 1 to 10 (1 = 'completely disagree', 10 = 'completely agree'), to what extent they agreed on. These measures included seven options that were under debate during the period of data collection as follows. (1) Increasing fines against citizens who did not follow social distancing and hygiene-related measures. (2) Using law enforcement agencies and the military for pervasive social control. (3) Introducing technologies to trace individual mobility. Promoting a downloadable smartphone application to trace contacts by making (4) its adoption voluntary with individual data anonymous for public authorities, (5) its adoption voluntary with personal identity revealed to public authorities, (6) its adoption mandatory with individual data anonymous for public authorities, or (7) its adoption mandatory with personal identity revealed to public authorities.

## 2.3. Sampling

The experiment was performed between 25 and 30 May 2020—i.e. at the end of the first wave of the pandemic and during the initial stages of 'Phase 2', but before the further loosening of mobility restrictions planned in mid-June. Participants included 1131 adults living in Lombardy and recruited through posts in pre-selected Facebook groups (on COVID-19 or regional issues) and through sponsored posts. To ensure independence of observations, all participants were required to comply with confidentiality and avoid discussing details related to the experiment during data collection. Comments on Facebook groups were monitored and deleted in case of any violation of confidentiality.

Figure 1 shows the geographical distribution of participants according to their residence and total population of each province. The most represented province in our sample was the Metropolitan City of Milan, which is also the most populated area, while the two light blue areas east of Milan (provinces of Bergamo and Brescia) were among the hardest hit areas in terms of excess mortality.

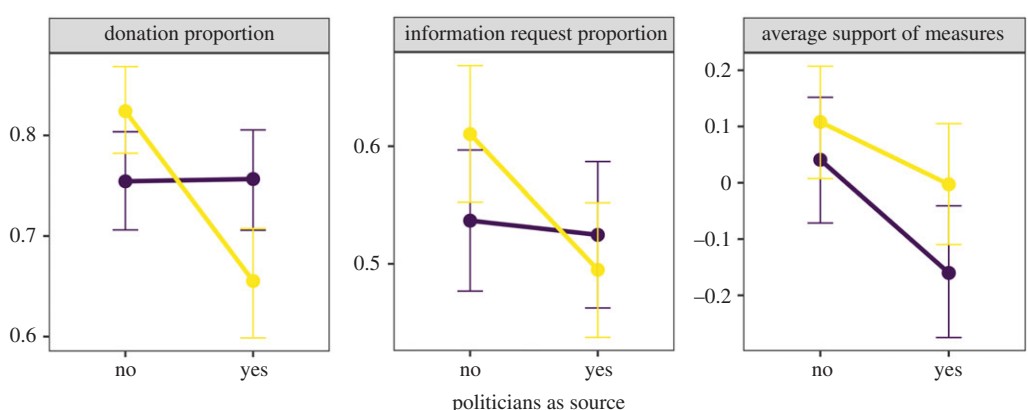

**Figure 2.** Mean decisions in treatments with politicians and/or scientific experts as sources of information; 95% confidence intervals obtained via bootstrap (1000 samples).

## 2.4. Data analysis

The analysis was performed using R 3.6.2 with the *brms* package to estimate the Bayesian models. Data (CovidExp.csv) and the scripts to replicate our analysis (SIanalysis.R and BrmsTables.R) are available on Harvard Dataverse at: https://doi.org/10.7910/DVN/UHNLHP.

## 3. Results

Of the 1131 participants, 75% were women, the median age was 44 years (s.d. = 14.3), and 56% had a university degree. Details on the sample demographics are reported in appendix A (table 8). In total, 74% of subjects chose to donate to a COVID-19-related charity while 54% asked for additional information on the measures mentioned in the experiment. Participants were required to express their agreement on seven individual statements, with different types of norm enforcement measures, including higher fines and more pervasive police control. Given that they consistently agreed across all these seven statements, we combined all responses into one single factor (details of factor analysis are presented in appendix A, table 13).

Figure 2 shows the means of our outcome measures per treatment and their confidence intervals. Results indicated that participants decided to donate, requested additional information, and supported more restrictive policy measures the most when these were recommended by scientific experts. In addition, the figure suggests a negative interaction between scientists and politicians, at least on donations and information requests. This means that, while information from scientists seemed to promote pro-social behaviour, mixing scientists and politicians led to even less cooperative attitudes than when politicians were the only source of information.

Tables 1 and 2 show Bayesian estimates of logistic regression models on the two main subject decisions: the donation and the request for further information on recommended measures. The first column of both tables shows estimates on the entire sample. Given that the COVID-19 mortality rate varies strongly with age and gender (i.e. deceased patients were mostly elderly and men), we also ran separate models based on these demographics, which are included in the other table columns. Appendix A also includes separate analyses regarding participants' educational level and political preferences (tables 9 and 10). Note that whenever we interpreted estimates here, we checked whether they were consistent across different demographic variables. If not mentioned explicitly, we found no qualitative difference across demographic groups.

Results in table 1 and 2 indicate that participants generally required more additional information regarding COVID-19 measures and donated more often to COVID-19-related charities when only scientists were the source of information. Furthermore, we found a negative interaction effect on both donations and information-seeking behaviour when information on measures was presented by a joint statement between politicians and scientific experts. It is worth noting that when information signals came from politicians *alone*, we did not find any consistent pure effect (tables 1–2). In general, the reported effects were stronger on older participants.

**Table 1.** Estimates, credible intervals and Bayes factors (for estimate > 0) of logistic regression models predicting whether participants donated.

| variable | statistic | overall | women | men | age ≤ 44 | age > 44 |
|---|---|---|---|---|---|---|
| intercept | Est. | 1.08 | 1.34 | 0.61 | 1.08 | 1.04 |
| | CI 95 | [0.37,v1.61] | [0.29,v2.86] | [−0.26, 1.7] | [0.1,v2.12] | [−0.56, 2.02] |
| | BF | 1000:1 | 55:1 | 18:1 | 47:1 | 8:1 |
| politicians | Est. | 0.01 | −0.13 | 0.41 | −0.34 | 0.59 |
| | CI 95 | [−0.36, 0.39] | [−0.55, 0.3] | [−0.43, 1.17] | [−0.85, 0.16] | [−0.05, 1.17] |
| | BF | 1:1 | 1:3 | 5:1 | 1:10 | 25:1 |
| scientists | Est. | 0.42 | 0.32 | 0.71 | 0.25 | 0.62 |
| | CI 95 | [0.01, 0.82] | [−0.14, 0.76] | [−0.02, 1.5] | [−0.27, 0.79] | [0.06,1.21] |
| | BF | 52:1 | 12:1 | 31:1 | 4:1 | 66:1 |
| Pol. × Sci. | Est. | −0.92 | −0.68 | −1.49 | −0.39 | −1.67 |
| | CI 95 | [−1.44, − 0.39] | [−1.3, − 0.09] | [−2.59, −0.44] | [−1.08, 0.34] | [−2.48, − 0.86] |
| | BF | 1:999 | 1:76 | 1:1000 | 1:6 | 1:1000 |

**Table 2.** Estimates, credible intervals and Bayes factors (for estimate > 0) of logistic regression models predicting whether participants requested extra information on measures included in the vignettes.

| variable | statistic | overall | women | men | age ≤ 44 | age > 44 |
|---|---|---|---|---|---|---|
| intercept | Est. | −0.04 | −0.02 | −0.13 | −0.09 | 0.01 |
| | CI 95 | [−0.15, 0.08] | [−0.14, 0.11] | [−0.36, 0.12] | [−0.25, 0.08] | [−0.14, 0.17] |
| | BF | 1:3 | 1:2 | 1:6 | 1:7 | 1:1 |
| politicians | Est. | 0.01 | −0.07 | 0.24 | 0.03 | −0.01 |
| | CI 95 | [−0.17, 0.18] | [−0.26, 0.12] | [−0.08, 0.56] | [−0.2, 0.26] | [−0.24, 0.21] |
| | BF | 1:1 | 1:3 | 12:1 | 1:1 | 1:1 |
| scientists | Est. | 0.2 | 0.15 | 0.38 | 0.39 | 0.02 |
| | CI 95 | [0.03, 0.36] | [−0.03, 0.32] | [0.06, 0.73] | [0.16, 0.62] | [−0.2, 0.22] |
| | BF | 142:1 | 15:1 | 124:1 | 999:1 | 1:1 |
| Pol. × Sci. | Est. | −0.25 | −0.13 | −0.65 | −0.51 | 0 |
| | CI 95 | [−0.49, − 0.01] | [−0.39, 0.13] | [−1.08, −0.19] | [−0.84, − 0.2] | [−0.31, 0.31] |
| | BF | 1:52 | 1:5 | 1:1000 | 1:1000 | 1:1 |

Table 3 shows our results on the level of agreement on norm enforcement measures to counteract a potential pandemic resurgence. We found that whenever politicians alone were the source of information, participants were less inclined to support these measures. Note that this effect was mainly driven by women and was most pronounced for younger participants, which would confirm recent findings on COVID-19 rates of compliance in other countries [10,11].

## 4. Discussion and conclusion

In many regions afflicted by the COVID-19 pandemic, such as Lombardy, the efficacy of restrictive measures to contain the COVID-19 pandemic required a high level of cooperation by millions of citizens. Press conferences, political announcements and TV interviews by public authorities were instrumental to inform and reassure the public. During this 'infodemic', politicians and scientific experts often went hand-in-hand urging the public to comply with regulations that, in most cases,

**Table 3.** Estimates, credible intervals and Bayes factors (for estimate > 0) of linear regression models predicting participants' support to measures.

| variable | statistic | overall | women | men | age ≤ 44 | age > 44 |
|---|---|---|---|---|---|---|
| intercept | Est. | 0.04 | 0.08 | −0.1 | 0.11 | −0.02 |
| | CI 95 | [−0.08, 0.15] | [−0.06, 0.2] | [−0.34, 0.13] | [−0.04, 0.26] | [−0.18, 0.13] |
| | BF | 3:1 | 8:1 | 1:4 | 14:1 | 1:2 |
| politicians | Est. | −0.2 | −0.26 | 0 | −0.25 | −0.16 |
| | CI 95 | [−0.36, −0.05] | [−0.43, −0.06] | [−0.33, 0.35] | [−0.46, −0.03] | [−0.39, 0.07] |
| | BF | 1:142 | 1:199 | 1:1 | 1:99 | 1:11 |
| scientists | Est. | 0.07 | 0.01 | 0.25 | −0.02 | 0.15 |
| | CI 95 | [−0.09, 0.22] | [−0.17, 0.21] | [−0.07, 0.57] | [−0.24, 0.19] | [−0.06, 0.36] |
| | BF | 5:1 | 1:1 | 14:1 | 1:1 | 10:1 |
| Pol. × Sci. | Est. | 0.09 | 0.19 | −0.19 | 0.22 | −0.02 |
| | CI 95 | [−0.12, 0.32] | [−0.08, 0.46] | [−0.67, 0.25] | [−0.1, 0.52] | [−0.32, 0.31] |
| | BF | 4:1 | 14:1 | 1:5 | 11:1 | 1:1 |

required a majority of less vulnerable citizens to pay a cost in terms of liberty and privacy infringements to benefit public health and more vulnerable subjects [2].

While in the case of Italy, norm compliance was probably influenced by fear during the so-called 'Phase 1', where the number of infections and deaths was considerable, understanding just how to ensure regulation enforcement is even more challenging now. Indeed, it is probable that the next pandemic waves will alternate spikes and drops of infection rates, so requiring adaptive regulations and patience by the public. The rise of street protests in various countries, with some episodes of social unrest when restrictions were announced in October and November 2020, suggests that compliance enforcement is still a key point of this global public health emergency [43].

Our results confirm that, at least in our experimental setting, individuals were sensitive to the source of information behind policy measures [12,17,18]. Interestingly, we were able to alter our participants' attitudes by minimal changes in the context of information [31,36,44]. Participants were more sensitive to pro-social motivations when the source of information was scientific experts *alone* without politicians. They were also less willing to accept more restrictive measures whenever these were recommended by politicians *alone*. Most notably, we found a decline in pro-social motivations whenever the same measures were presented by both politicians and scientific experts.

A possible explanation of this effect is that co-responsibility creates confusion in the information signal [34]. Due to public pressure for immediate policy responses and considering the usually misplaced expectations about the role of science and fundamental misunderstanding about the (un)certainty and experimental nature of scientific knowledge, it is probable that public emergencies can exacerbate confusion of public responsibility [45]. As suggested by Rosella *et al.* [34] in a policy analysis on the Pandemic H1N1 in Canada, the lack of clarification of roles and responsibilities between public decision-makers and experts could weaken public trust and compromise compliance. In the 24 h news cycle of globally connected societies, the interaction of 'politicized' scientists and 'science converted' politicians [46] can confuse citizens. This may be because: (i) scientific experts are induced to compete in an expertise set for political and media attention, thus reducing scientific advice to a merely symbolic or rhetorical discourse; and (ii) the legitimacy of public decisions by politicians is perceived as dependent on controversial scientific experts and so fundamentally unstable [21,47].

While political distrust is not new to Italy [38] and the COVID-19 pandemic has probably increased the credibility of scientific experts [48], the fact that public decisions, announcements and information were often shared by a policy/science liaison could generate confusion in the public about the source of public responsibility. While under 'Phase 1' stages, the high number of infections may have helped politicians to impose stricter measures, the possibility of enforcing social control during the long and complicated 'Phase 2' required more attention. On the other hand, regulation enforcement of social distancing and hygiene-related measures greatly depend on public behaviour when norm enforcement is difficult, the dynamics of infections shows peaks and oscillates, and people are tired and frustrated by months of personal sacrifice.

Furthermore, while this confusion can have immediate effects on the trust and credibility of politicians, the public role and exposure of scientific experts could backfire on the credibility of science in the longer run. As suggested by Stevens [49], although politicians can present public measures as a result of scientific advice, as if science were an 'apolitical and indisputable tablet of stone', this does not increase the legitimacy and trust of the institutions they represent. While in some countries such as Sweden it is expected, and in some cases even legally defined, that scientific experts take on a public role in certain circumstances, it is probable that in most countries the mutual role and boundaries between public decision-makers and scientific experts have never been clearly defined [50]. Although a more nuanced understanding of the interplay of institutions, norms and social behaviour is needed, especially in the context of a sequence of pandemic waves [51], our results suggest that public communication by experts and decision-makers can introduce information noise, which could be detrimental to enforce public cooperation and regulation compliance.

This said, our study also has certain limitations. The first is the lack of a representative random sample of Lombardy population. While this should be the gold standard of social research, it is difficult to achieve such a standard in behavioural online experiments aiming to examine a timely and urgent issue. In these cases, there is a trade-off between the need to collect data on people's responses to specific circumstances and the quality of the sample. This was the case of Italy during the first wave of the pandemic, with many activities limited or running at a reduced pace, which made it more difficult to use standard offline or online panels. Moreover, our experimental study aimed to test a cause–effect relationship between the source of information and the reaction of subjects, with priority given to internal validity against generalizability.

Another limitation is that participants were recruited through Facebook. While this ensured a good distribution of age, our sample was unbalanced in terms of gender and education, with a disproportionate number of women and highly educated subjects. On the one hand, this could show a higher propensity of this type of population to engage in scientific experiments. On the other, it is probable that the fact that we could not mention any reward in the recruitment Facebook message, given that the reward was part of the experimental manipulation itself, may have discouraged men from participating [52]. While previous research has shown gender differences in the level of norm compliance during the pandemic, with men showing lower rates [52], the fact that we did not find any crucial gender or education differences in the key factors in our observations, makes us confident that such a possible bias did not qualitatively affect our findings. However, given that the pandemic is unfolding with periods of restrictions and easing of lock-down measures, not entirely predicted when we designed our study, repeating our experiment with a representative sample of the population would help corroborate our findings. However, caveats aside, our study has shown the fruitfulness of running behavioural experiments online to understand responses of large populations to critical conditions *in the making*.

Finally, our findings further confirm the sensitivity of participants to information signals, which are key especially to public communication [53]. In our case, it was sufficient to add minor modifications in the vignettes to generate different responses by participants [36]. Furthermore, our findings suggest the importance of political communication as a means to promote trust regarding policy measures in public health emergencies [4]. While attention has been recently addressed on how political and science communication can increase a decentralized web governance that empowers people [54], the COVID-19 pandemic crisis has shown that traditional political institutions must also be concerned about how communication and information signals can influence public perception of decision-making roles and responsibility [53,55].

Ethics. The study was approved by the Ethics Committee of the University of Milan on 21 May 2020.

Data accessibility. The analysis was performed through R 3.6.2 with the brms package to estimate the Bayesian models. Data (CovidExp.csv) and the scripts to replicate our analysis (SIanalysis.R and BrmsTables.R) are available at https://doi.org/10.7910/DVN/UHNLHP.

Authors' contributions. All authors jointly designed the study and wrote the paper. M.F. coded the experiment and performed data analysis. F.B. and M.F. performed data collection.

Competing interests. We declare we have no competing interests.

Funding. No funding has been received for this article.

Acknowledgements. Our study was supported by the Linnaeus University Center for Data Intensive Sciences and Applications, DISA (lnu.se/disa) and the BEHAVE lab of the Department of Social and Political Sciences, University of Milan (http://behavelab.org/). We would like to thank Marco Pedrazzi and all members of the Ethics Committee of the University of Milan for useful remarks and suggestions on the experimental design. We would like to thank Robert Coates for the linguistic revision of the text.

# Appendix A

## A.1. Structure of the experiment

The experiment consisted of three blocks.

*Block 1*: The participants were randomly confronted with one of four vignettes (i.e. our experimental manipulation), informing on suggested measures to contain the COVID-19 pandemic during 'Phase 2'. The only difference between the four vignettes was the source of information regarding recommended measures, i.e. politicians, scientists or both. These differences were minimized in that groups differed only in whether statements in the vignette were introduced with statements of the kind 'According to [SOURCE]' or not. The source could either be *scientific experts*, *politicians* or *scientific experts and politicians*. After the vignette, participants were asked whether they wanted additional information regarding the statements in the vignette, which they could download after the experiment ended. This was to measure the effect of the source manipulation in the vignette on information-seeking behaviour and the response was used as one of the three outcome measurements in the analysis.

*Block 2*: Participants were confronted with seven statements regarding suggested measures to contain the COVID-19 pandemic which they were required to agree on. These measures were actually debated in the public media as possible counter-measures in the case of norm violation during 'Phase 2' and for social control, including the release of a governmental COVID-19 app to trace people's contacts via smartphone Bluetooth. Statements were as follows:

(1)  Those who do not respect norms on social distancing or don't use face masks should be sanctioned with high fines.
(2)  Law enforcement agencies and the military should be employed to enforce social distancing and the use of face masks in the streets and public places.
(3)   Everybody's mobility should be traced.
(4)  A *voluntarily* downloadable smartphone application, tracing contacts between infected and non-infected persons, storing *anonymous* data for public authorities.
(5)  A *voluntarily* downloadable smartphone application, tracing contacts between infected and non-infected persons, storing data for public authorities, including personal *identity*.
(6)  A *mandatory* smartphone application, tracing contacts between infected and non-infected persons, storing *anonymous* data for public authorities.
(7)  A *mandatory* smartphone application, tracing contacts between infected and non-infected persons, storing data for public authorities, including personal *identity*.

Furthermore, participants were informed that, as part of the experiment, they received a lottery ticket giving them a 1-in-50 chance to win a €50 Amazon voucher. Participants could choose to donate the €1 value of the lottery ticket to a charity linked to COVID-19. They were informed that this donation would be done by the experimenters in their name. To avoid path-dependency between the order in which statements and the donation decisions were presented, the order was randomized.

*Block 3*: Finally, participants were required to provide demographic information, were debriefed on the purpose of the experiment, were required not to discuss the content of the experiment for one week, and the lottery was resolved in the case of participants choosing to keep the lottery ticket.

## A.2. Instructions

*Disclaimer: The following is an English translation of the original questionnaire, which was administered in Italian. The original Italian instructions can be found at https://doi.org/10.7910/DVN/UHNLHP in the document entitled 'Protocollo.pdf'.*

### A.2.1. Welcome page

Welcome!

Thank you for your interest in our study.

If you grant us your informed consent, you will be presented a 15-minute long questionnaire, which is part of a research conducted by the University of Milan and the Linnaeus University (Växjö, Sweden).

Our aim is to study opinions on norms against COVID-19 contagion among residents in the Lombardy region.

Your participation is free and voluntary. If you choose to participate, you will nonetheless be able to leave at any moment. As a reward for your time, you will be able to join a lottery with a €50 worth Amazon voucher (1 voucher at stake every 50 participants).

The study has been approved by the Ethics Committee of the University of Milan. Participation to the study will be completely anonymous. Data will be anonymously stored as to comply to privacy protection laws (GDPR of the European Union). Data will not be accessible to anybody outside the research team, nor will it be transferred to third parties. The aim of this study is purely scientific.

### A.2.2. Vignette

Lombardy is still the hardest hit Italian region by COVID-19. Although anti-contagion norms have been relaxed during 'Phase 2', *[SOURCE]* the current situation remains dangerous and there is substantial risk of a new contagion wave. In order to avoid this, *[SOURCE]* social distancing, the use of face masks, and quarantine for suspected infected persons need to be kept in force. Furthermore, *[SOURCE]* people's mobility should be traced, in order to allow law enforcement to trace back infected people's contacts. *[SOURCE]* these measures are necessary to substantially decrease the risk of a new contagion wave.

**[SOURCE]** = {' ', 'According to politicians', 'According to scientists', 'According to politicians and scientists'}

### A.2.3. Outcome measures

(i) Would you like to obtain further information on the sources cited in the above text? [If yes] More information will be available at the end of the questionnaire.
(ii) On a scale from 1 to 10 (1 = 'completely disagree', 10 = 'completely agree'), to what extent do you agree with the following possible anti-contagion norms?
  a. Those who don't respect norms on social distancing or don't use face masks should be sanctioned with high fines.
  b. Law enforcement agencies and the military should be employed to enforce social distancing and the use of face masks in the streets and public places.
  c. Everybody's mobility should be traced.
(iii) On a scale from 1 to 10 (1 = 'completely disagree', 10 = 'completely agree'), to what extent do you agree with the following possible interventions?
  a. A *voluntarily* downloadable smartphone application, tracing contacts between infected and non-infected persons, storing *anonymous* data for public authorities.
  b. A *voluntarily* downloadable smartphone application, tracing contacts between infected and non-infected persons, storing data for public authorities, including personal *identity*.
  c. A *mandatory* smartphone application, tracing contacts between infected and non-infected persons, storing *anonymous* data for public authorities.
  d. A mandatory smartphone application, tracing contacts between infected and non-infected persons, storing data for public authorities, including personal *identity*.
(iv) To thank you for your participation, you will receive a ticket to join a lottery whose prize is a €50.00 worth Amazon voucher. Winning tickets will be randomly drawn every 50 participants. Alternatively, you can donate €1 to one of the following non-profit organizations engaged in contrasting the COVID-19 pandemic in Italy.

— ASST Fatebenefratelli Sacco
— Protezione Civile.

### A.2.4. Personal information

(i) Please report your age.
(ii) What is your gender?
  — Female
  — Male
  — I do not identify with any of the previous options

(iii) Please, select your highest education degree.
    — Primary education
    — Secondary education (first degree, It. *Scuola media inferiore*)
    — Secondary school (second degree, A-level)
    — University degree
(iv) What is your occupational status at the moment?
    — Dependent worker (private sector)
    — Dependent worker (public sector)
    — Retired
    — Self-employed
    — Unemployed
    — Other [please, complete]
(v) Please, select your residence municipality.
(vi) Many people position their political preferences within a spectrum between left and right. Where would you position yourself? Please, select a position between 0 (= extreme left) and 10 (extreme right).
(vii) Did you vote at the last Italian general elections in 2018?
(viii) Have you tested positive to SARS-CoV-2?
(ix) On a scale from 1 to 10 (1 = 'not at all', 10 = 'very much'), to what extent do you consider COVID-19 to be a threat for your health and that of your beloved ones?
(x) Has any of your closest relatives and friends developed COVID-19?

### A.2.5. Debriefing

Thank you for participating to this study. Our aim is to study the effect of political institutions and the scientific community on the attitudes of Lombardy residents. In order to avoid biasing other participants, it is very important that you refrain from discussing the content of this questionnaire with other potential subjects for at least two weeks, including commenting on social media (e.g. Facebook).

We would be glad to receive your feedback on the questionnaire, including criticism and suggestions. Once again, thank you for your participation.

## A.3. Supplementary analysis

### A.3.1. Frequentist analysis

Tables 4, 5, and 6 show regression estimates based on maximum likelihood, which have been designed to provide a frequentist replication of the Bayesian estimates included in the main text. Note that these results shows qualitative outcomes similar to those reported in the main text. These tables also provide the partial-$R^2$ value per estimate as a measure of effect strength of our experimental manipulation.

### A.3.2. Sample demographics

In the final block of the experiment, we included a battery of demographic questions. If not reported separately in the main text, we did not find any interactions between the treatment and these variables. In our sample, 75% were female. Figure 3 shows the age distribution of the sample.

Here 90% of participants declared they voted during the last national election. Figure 4 shows the entire spectrum of political preferences of participants. It indicates that our sample was dominated by left-leaning voters.

We also asked subjects whether they had been infected by the virus or knew anyone infected among their closest relatives or friends, and 34% of participants responded positively. Figure 5 shows that COVID-19 was clearly perceived as a threat by participants to their own health.

Table 7 shows the distribution of the sample across the provinces within Lombardy. Different provinces were more or less proportionally represented.

### A.3.3. Demographics across treatments

Participants were randomly assigned to the treatment and we found no significant differences of the demographic variables across treatments. Table 8 shows their distribution.

**Table 4.** Frequentist replication of table 1, predicting how much participants donated. Values in round brackets show standard errors; in square brackets partial-$R^2$.

|  | overall | women | men | age $\leq$ 44 | age > 44 |
|---|---|---|---|---|---|
| politicians | 0.012 | −0.126 | 0.423 | −0.337 | 0.513 |
|  | (0.199) | (0.233) | (0.389) | (0.268) | (0.312) |
|  | [0.000] | [0.000] | [0.004] | [0.003] | [0.005] |
| scientists | 0.422* | 0.316 | 0.720 | 0.249 | 0.600* |
|  | (0.207) | (0.243) | (0.403) | (0.286) | (0.302) |
|  | [0.004] | [0.002] | [0.012] | [0.002] | [0.007] |
| Pol. × Sci. | −0.915** | −0.671* | −1.498** | −0.368 | −1.580*** |
|  | (0.280) | (0.331) | (0.540) | (0.383) | (0.427) |
|  | [0.009] | [0.005] | [0.028] | [0.002] | [0.024] |
| intercept | 1.122*** | 1.299*** | 0.573* | 1.052*** | 1.192*** |
|  | (0.138) | (0.163) | (0.267) | (0.211) | (0.196) |
| N | 1131 | 855 | 272 | 554 | 574 |
| $R^2$ | 0.019 | 0.014 | 0.036 | 0.015 | 0.028 |

Note: *$p < 0.05$; **$p < 0.01$; ***$p < 0.001$

**Table 5.** Frequentist replication of table 2, predicting whether participants requested additional information regarding COVID-19. Values in round brackets show standard errors; in square brackets partial-$R^2$.

|  | overall | women | men | age $\leq$ 44 | age > 44 |
|---|---|---|---|---|---|
| politicians | 0.006 | −0.069 | 0.233 | 0.033 | −0.018 |
|  | (0.081) | (0.092) | (0.170) | (0.113) | (0.115) |
|  | [0.000] | [0.001] | [0.007] | [0.000] | [0.000] |
| scientists | 0.197* | 0.146 | 0.376* | 0.389*** | 0.017 |
|  | (0.079) | (0.089) | (0.169) | (0.113) | (0.110) |
|  | [0.006] | [0.003] | [0.018] | [0.021] | [0.000] |
| Pol. × Sci. | −0.252* | −0.129 | −0.642** | −0.510** | 0.003 |
|  | (0.113) | (0.129) | (0.232) | (0.159) | (0.159) |
|  | [0.004] | [0.001] | [0.028] | [0.018] | [0.000] |
| intercept | −0.039 | −0.016 | −0.123 | −0.091 | 0.010 |
|  | (0.056) | (0.063) | (0.121) | (0.079) | (0.079) |
| N | 1131 | 855 | 272 | 554 | 574 |
| $R^2$ | 0.010 | 0.008 | 0.031 | 0.037 | 0.000 |

Note: *$p < 0.05$; **$p < 0.01$; ***$p < 0.001$

### A.3.4. Treatment effects across demographics

Besides the demographic variables *age* and *sex* for which treatment effects were checked separately, we also checked for differences of the treatment effects in demographic groups based on their education and their self-positioning on a left–right political spectrum. For both demographics, we split the sample in two groups that were approximately of the same size. In the case of education, we split the sample on tertiary education or higher (55%) and the median political preference was 4 (above the participants fall in the *right* category). Tables 9 and 10 show that the effect of the source of information did not differ systematically for these factors (see tables 11 and 12).

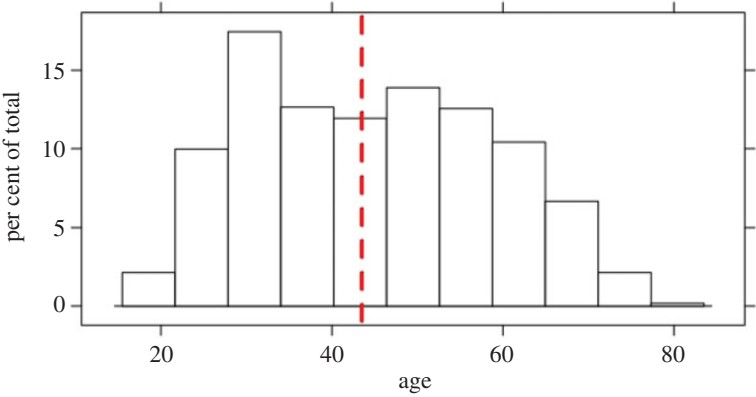

**Figure 3.** Age distribution in the sample. The red line indicates the median.

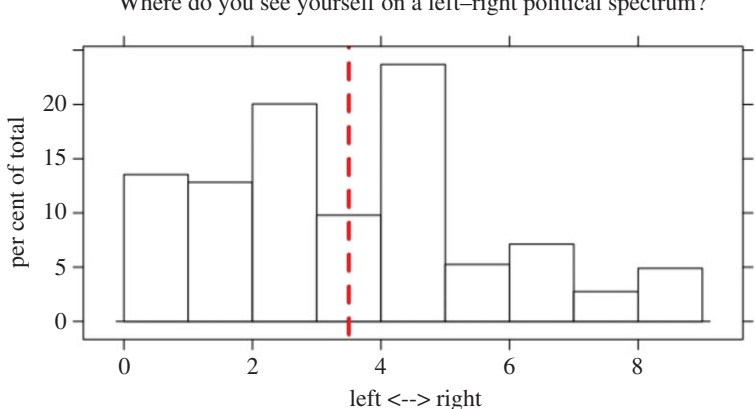

**Figure 4.** Distribution of political preferences in the sample. Red line indicates the median.

**Table 6.** Frequentist replication of table 3, predicting participants' support to measures. Values in round brackets show standard errors; in square brackets partial-$R^2$.

|                | overall     | women        | men     | age ≤ 44 | age > 44 |
| -------------- | ----------- | ------------ | ------- | -------- | -------- |
| politicians    | −0.201*     | −0.263**     | −0.005  | −0.251*  | −0.155   |
|                | (0.080)     | (0.091)      | (0.167) | (0.110)  | (0.116)  |
|                | [0.006]     | [0.010]      | [0.000] | [0.009]  | [0.003]  |
| scientists     | 0.067       | 0.008        | 0.245   | −0.025   | 0.149    |
|                | (0.078)     | (0.088)      | (0.165) | (0.110)  | (0.111)  |
|                | [0.001]     | [0.000]      | [0.008] | [0.000]  | [0.003]  |
| Pol. × Sci.    | 0.090       | 0.193        | −0.182  | 0.222    | −0.018   |
|                | (0.111)     | (0.128)      | (0.227) | (0.155)  | (0.160)  |
|                | [0.001]     | [0.003]      | [0.002] | [0.004]  | [0.000]  |
| intercept      | 0.041       | 0.079        | −0.101  | 0.110    | −0.025   |
|                | (0.055)     | (0.062)      | (0.119) | (0.077)  | (0.079)  |
| N              | 1131        | 855          | 272     | 554      | 574      |
| $R^2$          | 0.011       | 0.013        | 0.011   | 0.011    | 0.013    |

Note: *$p < 0.05$; **$p < 0.01$; ***$p < 0.001$

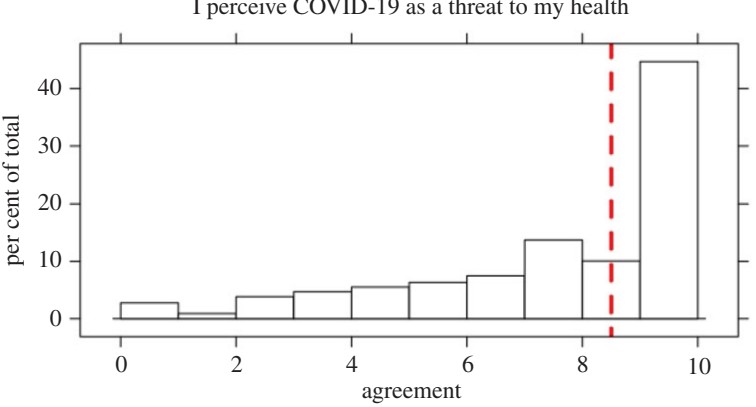

**Figure 5.** Distribution of COVID-19 threat perception in the sample. Red line indicates the median.

**Table 7.** Number of observations per province. Province population and excess mortality in March 2020 compared with 2015–2019 averages from [1]. Number of confirmed COVID-19 cases as 15 May 2020 from [56].

| province name | population (millions) | COVID-19 cases | excess mortality (%) | sample size |
|---|---|---|---|---|
| Bergamo | 1.114 | 12 371 | 571.3 | 77 |
| Brescia | 1.266 | 14 008 | 292.0 | 116 |
| Como | 0.599 | 3612 | 63.4 | 79 |
| Cremona | 0.359 | 6303 | 401.3 | 22 |
| Lecco | 0.337 | 2616 | 183.9 | 35 |
| Lodi | 0.230 | 3325 | 377.1 | 28 |
| Mantova | 0.412 | 3281 | 122.9 | 19 |
| Milano | 3.250 | 21 966 | 94.9 | 460 |
| Monza e Brianza | 0.874 | 5219 | 100.6 | 151 |
| Pavia | 0.546 | 4919 | 135.8 | 33 |
| Sondrio | 0.181 | 1339 | 77.6 | 16 |
| Varese | 0.891 | 3335 | 32.0 | 95 |

**Table 8.** Distribution of demographic variables across treatments.

| source | | age | | threat perception | | political pref. | | proportion in sample | | | |
|---|---|---|---|---|---|---|---|---|---|---|---|
| Pol. | Sci. | mean | s.d. | mean | s.d. | mean | s.d. | female | COVID in fam. | voted | high edu. |
| no | no | 45.17 | 14.44 | 6.93 | 2.58 | 4.01 | 2.21 | 0.79 | 0.27 | 0.88 | 0.55 |
| yes | no | 44.32 | 13.60 | 6.76 | 2.59 | 4.03 | 2.19 | 0.76 | 0.37 | 0.92 | 0.56 |
| no | yes | 45.67 | 14.58 | 7.25 | 2.26 | 3.94 | 2.12 | 0.77 | 0.35 | 0.90 | 0.55 |
| yes | yes | 43.70 | 13.36 | 7.04 | 2.44 | 4.22 | 2.15 | 0.71 | 0.38 | 0.88 | 0.57 |

### A.3.5. Factor loadings on measures

Participants were presented with seven questions on norm enforcement measures in Block 2 of the experiment. They were requested to agree on each measure on a 0–9 scale. Table 13 shows the corresponding factor loadings.

**Table 9.** Estimates, credible intervals and Bayes factors (for estimate > 0) of regression models predicting the effect of treatments on the three subject decisions in samples by level of education.

| Var. | Stat. | donation | | more info | | policy support | |
|---|---|---|---|---|---|---|---|
| | | low edu | high edu | low edu | high edu | low edu | high edu |
| intercept | Est. | 0.887 | 1.354 | −0.045 | −0.027 | −0.034 | 0.066 |
| | CI 95 | [0.382, 1.452] | [0.316, 2.185] | [−0.345, 0.224] | [−0.544, 0.589] | [−0.448, 0.25] | [−0.408, 0.375] |
| | BF | 2000:1 | 76:1 | 1:2 | 1:1 | 1:1 | 4:1 |
| Pol. | Est. | −0.062 | 0.093 | −0.006 | 0.009 | −0.158 | −0.228 |
| | CI 95 | [−0.628, 0.499] | [−0.472, 0.663] | [−0.238, 0.223] | [−0.207, 0.222] | [−0.411, 0.096] | [−0.434, −0.029] |
| | BF | 1:1 | 2:1 | 1:1 | 1:1 | 1:9 | 1:79 |
| Sci. | Est. | 0.9 | 0.022 | 0.16 | 0.226 | 0.136 | 0.021 |
| | CI 95 | [0.313, 1.513] | [−0.524, 0.573] | [−0.071, 0.395] | [0.022, 0.421] | [−0.093, 0.37] | [−0.188, 0.214] |
| | BF | 666:1 | 1:1 | 12:1 | 66:1 | 7:1 | 1:1 |
| Pol.×Sci. | Est. | −1.092 | −0.811 | −0.241 | −0.265 | 0.061 | 0.099 |
| | CI 95 | [−1.913, −0.262] | [−1.563, −0.077] | [−0.568, 0.078] | [−0.554, 0.019] | [−0.314, 0.381] | [−0.152, 0.376] |
| | BF | 1:285 | 1:73 | 1:13 | 1:29 | 2:1 | 3:1 |

**Table 10.** Estimates, credible intervals and Bayes factors (for estimate > 0) of regression models predicting the effect of treatments on the three subject decisions after splitting the sample by political preferences.

| Var. | Stat. | donation | | more info | | policy support | |
|---|---|---|---|---|---|---|---|
| | | left | right | left | right | left | right |
| intercept | Est. | 1.287 | 1.064 | −0.135 | 0.066 | 0.212 | −0.08 |
| | CI 95 | [−0.213, 2.728] | [0.477, 1.891] | [−0.455, 0.227] | [−0.204, 0.373] | [−0.097, 0.636] | [−0.516, 0.336] |
| | BF | 27:1 | 332:1 | 1:8 | 2:1 | 13:1 | 1:3 |
| Pol. | Est. | 0.187 | −0.17 | 0.155 | −0.141 | −0.32 | −0.141 |
| | CI 95 | [−0.387, 0.771] | [−0.682, 0.406] | [−0.058, 0.388] | [−0.364, 0.076] | [−0.544, − 0.101] | [−0.365, 0.084] |
| | BF | 3:1 | 1:3 | 12:1 | 1:8 | 1:499 | 1:8 |
| Sci. | Est. | 0.265 | 0.515 | 0.319 | 0.076 | −0.073 | 0.148 |
| | CI 95 | [−0.347, 0.866] | [−0.048, 1.067] | [0.107, 0.539] | [−0.139, 0.291] | [−0.28, 0.156] | [−0.065, 0.371] |
| | BF | 5:1 | 28:1 | 249:1 | 3:1 | 1:3 | 9:1 |
| Pol.×Sci. | Est. | −1.102 | −0.749 | −0.415 | −0.077 | 0.311 | −0.03 |
| | CI 95 | [−1.897, − 0.291] | [−1.552, 0.017] | [−0.74, − 0.101] | [−0.385, 0.235] | [0, 0.613] | [−0.356, 0.277] |
| | BF | 1:332 | 1:36 | 1:1999 | 1:2 | 38:1 | 1:1 |

**Table 11.** Frequentist replication of estimates in table 9, subsetting participants on the basis of their level of education. Values in round brackets show standard errors; in square brackets partial-$R^2$.

| | donation | | more info | | policy support | |
|---|---|---|---|---|---|---|
| | low edu | high edu | low edu | high edu | low edu | high edu |
| politicians | −0.051 | 0.086 | −0.008 | 0.017 | −0.162 | −0.233* |
| | (0.278) | (0.290) | (0.121) | (0.109) | (0.123) | (0.105) |
| | [0.000] | [0.000] | [0.000] | [0.000] | [0.003] | [0.008] |
| scientists | 0.903** | 0.027 | 0.160 | 0.227* | 0.132 | 0.015 |
| | (0.315) | (0.280) | (0.118) | (0.107) | (0.120) | (0.102) |
| | [0.019] | [0.000] | [0.004] | [0.007] | [0.002] | [0.000] |
| Pol. × Sci. | −1.104** | −0.803* | −0.240 | −0.265 | 0.070 | 0.109 |
| | (0.417) | (0.387) | (0.169) | (0.151) | (0.172) | (0.145) |
| | [0.014] | [0.007] | [0.004] | [0.005] | [0.000] | [0.001] |
| constant | 0.862*** | 1.360*** | −0.043 | −0.036 | −0.010 | 0.082 |
| | (0.193) | (0.212) | (0.083) | (0.076) | (0.085) | (0.073) |
| observations | 500 | 631 | 500 | 631 | 500 | 631 |
| $R^2$ | 0.036 | 0.018 | 0.009 | 0.011 | 0.012 | 0.011 |

Note: *$p < 0.05$; **$p < 0.01$; ***$p < 0.001$

**Table 12.** Frequentist replication of estimates in table 10, subsetting participants on basis of their political preferences. Values in round brackets show standard errors; in square brackets partial-$R^2$.

| | donation | | more info | | policy support | |
|---|---|---|---|---|---|---|
| | left | right | left | right | left | right |
| politicians | 0.200 | −0.164 | 0.151 | −0.123 | −0.309** | −0.117 |
| | (0.307) | (0.264) | (0.117) | (0.112) | (0.111) | (0.114) |
| | [0.001] | [0.001] | [0.003] | [0.002] | [0.015] | [0.002] |
| scientists | 0.274 | 0.531 | 0.316** | 0.088 | −0.062 | 0.164 |
| | (0.306) | (0.282) | (0.115) | (0.108) | (0.109) | (0.110) |
| | [0.002] | [0.007] | [0.014] | [0.001] | [0.001] | [0.004] |
| Pol. × sci. | −1.086* | −0.750* | −0.413* | −0.099 | 0.306 | −0.060 |
| | (0.422) | (0.378) | (0.165) | (0.155) | (0.156) | (0.157) |
| | [0.012] | [0.007] | [0.012] | [0.001] | [0.007] | [0.000] |
| constant | 1.255*** | 1.023*** | −0.135 | 0.041 | 0.190* | −0.084 |
| | (0.219) | (0.182) | (0.083) | (0.076) | (0.079) | (0.077) |
| observations | 527 | 603 | 527 | 603 | 527 | 603 |
| $R^2$ | 0.020 | 0.022 | 0.016 | 0.009 | 0.017 | 0.010 |

Note: *$p < 0.05$; **$p < 0.01$; ***$p < 0.001$

## A.3.6. Power analysis

Table 14 shows the result of a power analysis that was run to determine the sample size of the population before the experiment. Simulations were based on the assumptions that in our baseline treatment 50% of the participants would request additional information/donate and that donations in general would follow a binomial distribution. We simulated different degrees of differences between baseline and treatment groups. Given that our treatments manipulation explained 2% of unique variance and a

**Table 13.** Loadings and proportion of explained variance of the level of support to measures. The complete list of the measures can be found under Block 2.

| measures | agreement |
| --- | --- |
| Item 1 | 0.50 |
| Item 2 | 0.48 |
| Item 3 | 0.81 |
| Item 4 | 0.55 |
| Item 5 | 0.68 |
| Item 6 | 0.81 |
| Item 7 | 0.80 |
| explained variance | 0.46 |

**Table 14.** Power analysis for the detection of treatment effects on the donation decision and information request.

| N per treatment | explained variance | power |
| --- | --- | --- |
| 250 | 0.01 | 0.646 |
| 250 | 0.02 | 0.885 |
| 250 | 0.05 | 1.000 |
| 500 | 0.01 | 0.893 |
| 500 | 0.02 | 0.996 |
| 500 | 0.05 | 1.000 |

**Table 15.** Power analysis for the detection of treatment effects on the policy support.

| N per treatment | explained variance | power |
| --- | --- | --- |
| 250 | 0.01 | 0.624 |
| 250 | 0.02 | 0.913 |
| 250 | 0.05 | 0.999 |
| 500 | 0.01 | 0.900 |
| 500 | 0.02 | 0.996 |
| 500 | 0.05 | 1.000 |

significance threshold of 0.05, 250 participants per treatment would lead to a statistical power greater than 0.8. We thus aimed for 250–300 participants per treatment.

Table 15 shows a similar power analysis for the detection of treatment effects on the policy support (assuming a normal distribution of policy support). The script for the simulations is available at https://doi.org/10.7910/DVN/UHNLHP.

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
