## [Peer Review File · Royal Society Open Science]

Review History

RSOS-201310.R0 (Original submission)

Review form: Reviewer 1

Is the manuscript scientifically sound in its present form?

Yes

Are the interpretations and conclusions justified by the results?

Yes

Is the language acceptable?

Yes

Do you have any ethical concerns with this paper?

No

Have you any concerns about statistical analyses in this paper?

Yes

Recommendation?

Major revision is needed (please make suggestions in comments)

Comments to the Author(s)

This paper describes an online experiment where a vignette is communicated to participants. This vignette includes information on Lombardy's problem with covid-19, the current risks, suggested interventions to manage the virus, and finally a statement the interventions are effective at reducing the risks. Participants are randomly assigned to a group that receives information on the source of this information: scientists, politicians, both, or neither (control group). The study appears to be conducted well, although a more detailed methods section is needed to provide the reader with the necessary information. The results are interesting and relevant, and although have been researched extensively in other fields, this provides new information as it is applied to covid. The results could have value to researchers and policy makers.

The key limitation of the study is that the authors did not recruit a representative sample, however the authors address this in the conclusion so no change is needed. Although I disagree with their explanation - it is not necessarily difficult to achieve quickly. Plenty of research agencies offer fast, nationally representative samples, although this may be different for Lombardy/Italy. The barrier is normally cost. There are also several points where the choice of words is either misleading/inaccurate or unclear. Specific points are detailed below.

Specific comments:

1. Language: I would recommend edits to some of the language throughout to make it clearer or more appropriate. Some examples:

Some of the language is not appropriate for a scientific article. For example, describing lock-down and social distancing as "draconian". I recommend removing this word

Some sentences I had to read several times to understand. For example, p3, line 49. Recommend re-wording the sentence: "This has led to a consensual understanding of the main factors leading to large-scale cooperation under conditions of strategic interdependence". Can this be made clearer?

In methods (a) (line 20), I would change "randomly confronted with one of four vignettes" to "randomly allocated to a group that contains one of four vignettes"

"trust and legitimacy" I found it hard to read and understand the many sentences that included "trust and legitimacy" added in, when it does not need to be there. For example: see p8, line 51:

"Our results confirm that individuals are sensitive to the source of trust and public legitimacy of policy measures". I'm sure exactly what you are trying to say here, part of the problem is that it's not clear, but also it may be inaccurate. If you are saying that your results confirm that individuals are sensitive to the source of policy measures, then I agree, and I think this is a much clearer way of describing your results. Adding the words "of trust and public legitimacy of" makes the sentence less clear and potentially misleading as you didn't measure trust of perceived/actual legitimacy. Rewording the many sentences to describe the "source of information" is much clearer than the "source of trust and legitimacy". Because ultimately what you are describing is a source credibility effect: the source of information, either politicians, scientists or both.

P8, line 55. "We found that individuals are more cooperative...". The authors didn't measure cooperation, which in this context I would define as working with others to achieve some covid-related goal. It could also mean cooperating with (adhering to) the government guidelines? However, the authors don't measure either of these things. Please be clearer about which outcomes you are referring to.

2. Measures: Although it is great to see the materials are fully open and accessible, there needs to be further clarification in the paper itself to help the reader understand what you have done. In particular, there should be a section in the methods, subtitle: Measures (or similar), where you can describe the outcomes that you have measured. For example, it appears you have

3 outcomes: donating to covid charity, requesting further info, and support for covid measures. Please add these three in with key details, including: where the measures came from (if you adapted them from existing research), how the variables were measured, how the measures were scored (eg On a scale from 1 to 10 (1 = "completely disagree", 10 = "completely agree")), etc. You can then remove the details in the second paragraph of the results section

3. Randomisation: In the methods section (a), The authors should add in how the participants were randomised. This is recommended by reporting guidelines for randomised studies eg consort. Presumably in this study the answer is the software that was used to host the questionnaire?

4. P9, line 43, you state that covid-19 has probably increased the credibility of scientific experts. Citation is needed for this claim. It's an interesting point, but I don't know of any evidence for this claim. If none exists, then this needs rewording

5. Questionnaire: It's great to see the questionnaire and other materials in English (for me and fellow English readers). However, given that the original was written in Italian, the Italian version should also be made available, either as an appendix or linked to an online repository.

6. The authors include interaction effects in the models and report that there are interaction effects on two of the three outcomes. The authors present three graphs in Figure 2 as a way of visualising the interaction term, but they don't appear to interpret these within the results section. All that is said is that there is a "negative interaction" but what does that mean in the current context? More detail of the interactions are needed to fully understand these results. The discussion will likely need updating following these change too.

Review form: Reviewer 2

Is the manuscript scientifically sound in its present form?

Yes

Are the interpretations and conclusions justified by the results?

Yes

Is the language acceptable?

Yes

Do you have any ethical concerns with this paper?

No

Have you any concerns about statistical analyses in this paper?

No

Recommendation?

Accept with minor revision (please list in comments)

Comments to the Author(s)

Abstract

1. Reconsider the use of the phrase 'non-clinical measures'. The WHO have been using the term 'public health measures'
2. In a non-clinical trial setting, should it be 'effectiveness' rather than 'efficacy'

Introduction

3. I would suggest the authors refrain from using the term 'draconian'
4. The introduction is written in a way that suggests that the pandemic is over. I would encourage the reframing of the text so that the messages coming across support the notion that they are useful now, not just when things are finished.
5. Again the sentiment of this sentence is not correct; 'Such norms applied to everyone, including the majority of the population with zero or minimal risk of suffering serious harm from the virus'. It is critical that compliance is high and that public health measures are equitable etc. Beyond this, there is of course the issue of asymptomatic transmission.
6. Currently the introduction is very long. Consider moving the section on roles/responsibility (page 4, section 17-40) into the discussion or tighten up the paragraphs. The opening paragraph could also be reduced to a couple of sentences, as the epidemiology/impact of the pandemic is well documented now.

Review form: Reviewer 3

Is the manuscript scientifically sound in its present form?

No

Are the interpretations and conclusions justified by the results?

No

Is the language acceptable?

Yes

Do you have any ethical concerns with this paper?

No

Have you any concerns about statistical analyses in this paper?

Yes

Recommendation?

Major revision is needed (please make suggestions in comments)

Comments to the Author(s)

Please see the attached file (Appendix A) for detailed comments.

Decision letter (RSOS-201310.R0)

Dear Dr Farjam

The Editors assigned to your paper RSOS-201310 "Dangerous liaisons during the COVID-19 pandemic: An online experiment on the effect of role ambiguity on public policy" have now received comments from reviewers and would like you to revise the paper in accordance with the

reviewer comments and any comments from the Editors. Please note this decision does not guarantee eventual acceptance.

Please submit your revised manuscript and required files (see below) no later than 21 days from today's (ie 12-Nov-2020) date. Note: the ScholarOne system will 'lock' if submission of the revision is attempted 21 or more days after the deadline. If you do not think you will be able to meet this deadline please contact the editorial office immediately.

on behalf of Dr Christina Demski (Associate Editor) and Essi Viding (Subject Editor)
openscience@royalsociety.org

Associate Editor Comments to Author (Dr Christina Demski):

All three reviewers see merit in this manuscript, although at least two of the reviewers would like to see some major revisions including more detail and description in the introduction, method and discussion. I would encourage you to engage with all the comments, make appropriate changes and/or provide responses to their questions and suggestions.

Reviewer comments to Author:

Reviewer: 1
Comments to the Author(s)

This paper describes an online experiment where a vignette is communicated to participants. This vignette includes information on Lombardy's problem with covid-19, the current risks, suggested interventions to manage the virus, and finally a statement the interventions are effective at

reducing the risks. Participants are randomly assigned to a group that receives information on the source of this information: scientists, politicians, both, or neither (control group). The study appears to be conducted well, although a more detailed methods section is needed to provide the reader with the necessary information. The results are interesting and relevant, and although have been researched extensively in other fields, this provides new information as it is applied to covid. The results could have value to researchers and policy makers.

The key limitation of the study is that the authors did not recruit a representative sample, however the authors address this in the conclusion so no change is needed. Although I disagree with their explanation – it is not necessarily difficult to achieve quickly. Plenty of research agencies offer fast, nationally representative samples, although this may be different for Lombardy/Italy. The barrier is normally cost. There are also several points where the choice of words is either misleading/inaccurate or unclear. Specific points are detailed below.

Specific comments:

1. Language: I would recommend edits to some of the language throughout to make it clearer or more appropriate. Some examples:

Some of the language is not appropriate for a scientific article. For example, describing lock-down and social distancing as “draconian”. I recommend removing this word

Some sentences I had to read several times to understand. For example, p3, line 49. Recommend re-wording the sentence: “This has led to a consensual understanding of the main factors leading to large-scale cooperation under conditions of strategic interdependence”. Can this be made clearer?

In methods (a) (line 20), I would change “randomly confronted with one of four vignettes” to “randomly allocated to a group that contains one of four vignettes”

“trust and legitimacy” I found it hard to read and understand the many sentences that included “trust and legitimacy” added in, when it does not need to be there. For example: see p8, line 51: “Our results confirm that individuals are sensitive to the source of trust and public legitimacy of policy measures”. I’m sure exactly what you are trying to say here, part of the problem is that it’s not clear, but also it may be inaccurate. If you are saying that your results confirm that individuals are sensitive to the source of policy measures, then I agree, and I think this is a much clearer way of describing your results. Adding the words “of trust and public legitimacy of” makes the sentence less clear and potentially misleading as you didn’t measure trust of perceived/actual legitimacy. Rewording the many sentences to describe the “source of information” is much clearer than the “source of trust and legitimacy”. Because ultimately what you are describing is a source credibility effect: the source of information, either politicians, scientists or both.

P8, line 55. “We found that individuals are more cooperative...”. The authors didn’t measure cooperation, which in this context I would define as working with others to achieve some covid-related goal. It could also mean cooperating with (adhering to) the government guidelines? However, the authors don’t measure either of these things. Please be clearer about which outcomes you are referring to.

2. Measures: Although it is great to see the materials are fully open and accessible, there needs to be further clarification in the paper itself to help the reader understand what you have done. In particular, there should be a section in the methods, subtitle: Measures (or similar), where you can describe the outcomes that you have measured. For example, it appears you have 3 outcomes: donating to covid charity, requesting further info, and support for covid measures. Please add these three in with key details, including: where the measures came from (if you adapted them from existing research), how the variables were measured, how the measures were scored (eg On a scale from 1 to 10 (1 = “completely disagree”, 10 = “completely agree”)), etc. You can then remove the details in the second paragraph of the results section

3. Randomisation: In the methods section (a), The authors should add in how the participants were randomised. This is recommended by reporting guidelines for randomised studies eg

consort. Presumably in this study the answer is the software that was used to host the questionnaire?

4. P9, line 43, you state that covid-19 has probably increased the credibility of scientific experts. Citation is needed for this claim. It's an interesting point, but I don't know of any evidence for this claim. If none exists, then this needs rewording

5. Questionnaire: It's great to see the questionnaire and other materials in English (for me and fellow English readers). However, given that the original was written in Italian, the Italian version should also be made available, either as an appendix or linked to an online repository.

6. The authors include interaction effects in the models and report that there are interaction effects on two of the three outcomes. The authors present three graphs in Figure 2 as a way of visualising the interaction term, but they don't appear to interpret these within the results section. All that is said is that there is a "negative interaction" but what does that mean in the current context? More detail of the interactions are needed to fully understand these results. The discussion will likely need updating following these change too.

Reviewer: 2

Comments to the Author(s)

Abstract

1. Reconsider the use of the phrase 'non-clinical measures'. The WHO have been using the term 'public health measures'

2. In a non-clinical trial setting, should it be 'effectiveness' rather than 'efficacy'

Introduction

3. I would suggest the authors refrain from using the term 'draconian'

4. The introduction is written in a way that suggests that the pandemic is over. I would encourage the reframing of the text so that the messages coming across support the notion that they are useful now, not just when things are finished.

5. Again the sentiment of this sentence is not correct; 'Such norms applied to everyone, including the majority of the population with zero or minimal risk of suffering serious harm from the virus'. It is critical that compliance is high and that public health measures are equitable etc. Beyond this, there is of course the issue of asymptomatic transmission.

6. Currently the introduction is very long. Consider moving the section on roles/responsibility (page 4, section 17-40) into the discussion or tighten up the paragraphs. The opening paragraph could also be reduced to a couple of sentences, as the epidemiology/impact of the pandemic is well documented now.

Reviewer: 3

Comments to the Author(s)

Please see the attached file for detailed comments.

===PREPARING YOUR MANUSCRIPT===

===PREPARING YOUR REVISION IN SCHOLARONE===

- If you are providing image files for potential cover images, please upload these at this step, and inform the editorial office you have done so. You must hold the copyright to any image provided.
- A copy of your point-by-point response to referees and Editors. This will expedite the preparation of your proof.

- Ensure that your data access statement meets the requirements at <https://royalsociety.org/journals/authors/author-guidelines/#data>. You should ensure that you cite the dataset in your reference list. If you have deposited data etc in the Dryad repository, please include both the 'For publication' link and 'For review' link at this stage.
- If you are requesting an article processing charge waiver, you must select the relevant waiver option (if requesting a discretionary waiver, the form should have been uploaded at Step 3 'File upload' above).
- If you have uploaded ESM files, please ensure you follow the guidance at <https://royalsociety.org/journals/authors/author-guidelines/#supplementary-material> to include a suitable title and informative caption. An example of appropriate titling and captioning may be found at https://figshare.com/articles/Table_S2_from_Is_there_a_trade-off_between_peak_performance_and_performance_breadth_across_temperatures_for_aerobic_scope_in_teleost_fishes_/3843624.

Author's Response to Decision Letter for (RSOS-201310.R0)

See Appendix B.

RSOS-201310.R1 (Revision)

Review form: Reviewer 1

Is the manuscript scientifically sound in its present form?

Yes

Are the interpretations and conclusions justified by the results?

Yes

Is the language acceptable?

Yes

Do you have any ethical concerns with this paper?

No

Have you any concerns about statistical analyses in this paper?

No

Recommendation?

Accept with minor revision (please list in comments)

Comments to the Author(s)

I'd like to thank the authors for putting so much work into revising their paper following the feedback from two other reviewers and myself. Overall, I am pleased with the changes that have been made however I do still disagree with the authors use of the term "cooperate" which I still am not convinced that you measure.

The outcomes are: 1) donations to a charity (which I agree with the authors can be called pro-social behaviour 2) requesting more information (not cooperation), and 3) support for policies (an attitude - not cooperation). I think that authors are trying to make the point that by supporting the policies they are cooperating with them. However, attitudes are not behaviours. The gap between intentions and behaviours is large, and they do not even measure intentions. It would be acceptable to hypothesise about cooperation (actual behaviour) in the discussion, but not to actually state "participants were more cooperative". I think that more precise language should be used throughout the paper to better describe the outcomes that were measured.

Review form: Reviewer 3

Is the manuscript scientifically sound in its present form?

Yes

Are the interpretations and conclusions justified by the results?

Yes

Is the language acceptable?

Yes

Do you have any ethical concerns with this paper?

No

Have you any concerns about statistical analyses in this paper?

No

Recommendation?

Accept as is

Comments to the Author(s)

Thank you for addressing the comments so diligently. Much improved. Enjoyed reading the updated version of the manuscript.

Decision letter (RSOS-201310.R1)

Dear Dr Farjam

On behalf of the Editors, we are pleased to inform you that your Manuscript RSOS-201310.R1 "Dangerous liaisons: An online experiment on the role of scientific experts and politicians in ensuring public support for anti-COVID measures" has been accepted for publication in Royal Society Open Science subject to minor revision in accordance with the referees' reports. Please find the referees' comments along with any feedback from the Editors below my signature.

Please submit your revised manuscript and required files (see below) no later than 7 days from today's (ie 23-Feb-2021) date. Note: the ScholarOne system will 'lock' if submission of the revision is attempted 7 or more days after the deadline. If you do not think you will be able to meet this deadline please contact the editorial office immediately.

on behalf of Prof Essi Viding (Subject Editor)
openscience@royalsociety.org

Associate Editor Comments to Author:

We apologise for the delay in completing the review of your revision, but hope the comments from the referees are both reassuring and helpful in guiding a final revision of the manuscript.

Reviewer comments to Author:

Reviewer: 3

Comments to the Author(s)

Thank you for addressing the comments so diligently. Much improved. Enjoyed reading the updated version of the manuscript.

Reviewer: 1

Comments to the Author(s)

I'd like to thank the authors for putting so much work into revising their paper following the feedback from two other reviewers and myself. Overall, I am pleased with the changes that have

been made however I do still disagree with the authors use of the term “cooperate” which I still am not convinced that you measure.

The outcomes are: 1) donations to a charity (which I agree with the authors can be called pro-social behaviour 2) requesting more information (not cooperation), and 3) support for policies (an attitude - not cooperation). I think that authors are trying to make the point that by supporting the policies they are cooperating with them. However, attitudes are not behaviours. The gap between intentions and behaviours is large, and they do not even measure intentions. It would be acceptable to hypothesise about cooperation (actual behaviour) in the discussion, but not to actually state “participants were more cooperative”. I think that more precise language should be used throughout the paper to better describe the outcomes that were measured.

===PREPARING YOUR MANUSCRIPT===

===PREPARING YOUR REVISION IN SCHOLARONE===

Author's Response to Decision Letter for (RSOS-201310.R1)

See Appendix C.

Decision letter (RSOS-201310.R2)

Dear Dr Farjam,

It is a pleasure to accept your manuscript entitled "Dangerous liaisons: An online experiment on the role of scientific experts and politicians in ensuring public support for anti-COVID measures" in its current form for publication in Royal Society Open Science.

COVID-19 rapid publication process:

We are taking steps to expedite the publication of research relevant to the pandemic. If you wish, you can opt to have your paper published as soon as it is ready, rather than waiting for it to be published the scheduled Wednesday.

This means your paper will not be included in the weekly media round-up which the Society sends to journalists ahead of publication. However, it will still appear in the COVID-19 Publishing Collection which journalists will be directed to each week (<https://royalsocietypublishing.org/topic/special-collections/novel-coronavirus-outbreak>).

If you wish to have your paper considered for immediate publication, or to discuss further, please notify openscience_proofs@royalsociety.org and press@royalsociety.org when you respond to this email.

Best regards,
Lianne Parkhouse
Editorial Coordinator

on behalf of Professor Essi Viding (Subject Editor)
openscience@royalsociety.org

Appendix A

Dangerous liaisons during the COVID-19 pandemic: An online experiment on the effect of role ambiguity on public policy

- Review comments

Page 2:

Line 21: What is an emergency norm? The term should be explained here I think.

Line 27: "When strict measures are announced and attempted without having strategies to enforce them, which contemplate efficient communication, public credibility of public decision-makers, and attention to trust formation"

Enforce here sounds like a top down approach of control and preventing of 'rule breaking'. However, the examples seem to point more at internal motivational factors, such as credibility and trust, that relate to compliance. This section should be rewritten to rectify this.

Line 29: Language/grammar issue "In order to record public health appliance with norms". I would suggest having the article proofread by a native speaker.

Line 39. The authors refer to the covid prevention measures as 'norms'. I think caution is warranted here. Social norms are an established social psychological construct. A government announcement or creation or enforcement of rules does not necessarily make those rules social 'norms'. I think the authors should discuss this point more deeply how exactly they justify the use of the term 'norms' here or rewrite.

Page 3:

Line 7: "promote large-scale cooperation on norm compliance". Again, I am not understanding clearly why the authors are talking about 'norm compliance' here. It needs to be established what the norm is. A solid definition of the terminology in general and specifically what is meant here is needed I think. Otherwise it is not clear what is meant by 'cooperation on norm compliance'.

I am not commenting on any further use of the term 'norm', as the authors are carrying this theme through (e.g. talking about norm enforcement etc.). In the rewriting process the whole manuscript should be adjusted accordingly.

Page 4, Methods

Could the authors elaborate on their hypotheses regarding information seeking behaviour? How is that related to trust in the source? I could see the argument going both ways. If people trust the source more they look for more info, possibly to educate themselves. But they could also be interested if they don't trust it, just to find out more on it and learn more about potential 'misinformation'. So I do not see a clear direction of the effect here.

It is unclear at what point the information seeking behaviour was measured. The text said 'at the end they were presented with the opportunity to seek more'. The methods section did not specify this further. It ends by saying the lottery was resolved. Was the information seeking measure taken after that or before? If after, I could see there being a bias from people for instance not having won the lottery and thus being in a more negative emotional state which in turn could have influenced their decision to seek further info or not. Emotions, especially positive ones can enhance prosocial

behaviour. So I see a risk of bias here. Could the authors elaborate on this please and detail how they avoided such bias and present relevant statistics? Thank you.

I see a potential confound and bias in their sampling. They used facebook groups and posts on covid 19 and regional issues. It is quite a specific sub population of users who are a) on the internet in times of a pandemic, b) on facebook, c) active on covid and regional groups, d) willing to participate in this research. Yet, the language that the authors use to talk about their results suggests that they are trying to provide insights on the general human relationship to sources of information and trust with regards to politicians and scientists. I am not sure if such broad claims can be made based on the available data. The authors should comment on this and state clearly how they have resolved such biases. It would be ideal if the authors could replicate their study with a different more representative and broad sample. If not, I think it would help to adjust the language and rewrite the claims of the paper.

A power analysis is missing. How do the authors justify their sample? How did they determine that it gives them enough power to detect the effects of interest? I would ask the authors to run a power analysis and detail exactly how they justify estimated effect size and ensuing sample size.

Figure 2: I do not understand the figure fully. The experiment had three groups, politicians only, scientists only and both. I don't understand why the graphs show politicians yes no on the x axis and scientists yes/no as some kind of interaction type plot. I think it would be clearer if the plot displayed the three groups next to each other.

Page 6: The authors run quite a few analyses, checking many different variables, e.g. education, political attitude and more, yet no rationale is given for why they are doing it. I see a mild danger of multiple comparison testing and potential for false positives due to the amount of analyses conducted. I would like to see a power analysis that provides back up that the conducted tests are warranted and reliable.

Further on page 6: The authors report an interaction analysis. Here a power justification is even more crucial as necessary sample size to run an interaction is significantly larger compared to main effects investigations, e.g. investigating differences between groups alone. Without a power analysis I am not sure that the reliability and soundness of the results can be adequately determined.

Did the authors preregister their hypotheses? That would give further back up to their analysis approach of testing many different models. It currently seems a bit arbitrary as to the conducted and reported tests.

I would like to see effect sizes reported as well. Without effect sizes it is difficult to judge the relevance of the presented findings.

I appreciate the use of Bayesian statistics for data analysis. However, I would like to see additional analysis using frequentist statistics to see whether results match. Bayesian stats heavily depend on set priors, so it would be a good robustness check to perform frequentist statistics to see whether the found effects hold. I see the combination of statistical approaches as a good check of the soundness of reported findings.

Discussion

In the discussion the authors claim that their result 'further confirm prospect theory'. I am not quite seeing how this is the case. Prospect theory did not play a major role in the introduction.

Furthermore, framing in prospect theory is most prominently about gains and losses, not about sources. If the authors have a different interpretation in mind this should be made more clear here with relevant references.

The authors suggest that their findings “suggest the importance of political communication as a means to promote trust and public legitimacy of policy measures”. I can’t quite see how this statement follows from the experiment and the data presented. What do the authors mean by political communication? If anything their results show that scientist focused communication might be more beneficial. Again, this point should be made clearer.

Last sentence: “Our findings suggest that confronted with uncertainty and global crisis, people would appreciate to reduce public ambiguity regarding roles, responsibility and decision-making”. I can’t see how the study results support this claim. The study is not really about uncertainty. There is no uncertainty presented to people; it is about donating, policy support, information seeking. And I don’t see how any of the measures speak to preferences of ambiguity reduction regarding roles and responsibilities. The results show that people might trust communication from scientists more. But the authors did not test how people react to messages that are for instance ambiguous in terms of their sources.

Appendix:

“Would you like to obtain further information on the sources cited in the above text? [If yes] More information will be available at the end of the questionnaire.”

“Would you like to obtain further information on the possible interventions cited in the above question? [If yes] More information will be available at the end of the questionnaire.”

This suggests that this measure was collected twice? If so, when and why? And which data was used for the presented analyses and results? The authors should clarify this.

Title:

Lastly, I believe the title is not quite appropriate for the manuscript. It seems to somewhat inaccurately represent the findings. I don’t see an aspect of ‘danger’ in their findings. The results show that levels of donating and information seeking vary by source. Is there an element of negative emotional consequences involved? If so, this should be reported. My point is that an absence of e.g. ‘good’ behaviour does not equal an occurrence of ‘negative’ behaviour into the other direction.

Furthermore, I don’t see how the paper is about role ambiguity. It is about different groups, but there is no ambiguity regarding roles involved as roles are clearly stated.

Lastly, the measures don’t test public policy specifically; they test donation behaviour and information seeking and support of policies. I do not think this can be called effect on ‘public policy’.

The title of the submitted manuscript does not match the title in the data repository. This is odd.

Appendix B

Response letter for Manuscript RSOS-201310

Dear RSOS editor,

we would like to thank the reviewers for their helpful and valuable comments and express our gratitude to the editor for the chance to revise our manuscript "Dangerous liaisons during the COVID-19 pandemic: An online experiment on the effect of role ambiguity on public policy". As you can see, the manuscript has been extensively revised, starting from the title and abstract. We also have refined the background section to reflect the right remark of referees that the theoretical focus was focused on norms and so could induce misinterpretation of our design. This required us to change various references, which now are more concentrated on regulation enforcements. Given that time has passed since our original submissions, we also have update the presentation and discussion of our findings to the current situation of the pandemic so that insights can have value also in view of the next waves of the pandemic and the probably sequences of restrictive regulations/gradual easing of measures, which will accompany the next months in various countries. This is to say that we believe that our findings can still contribute to the current debate about the cooperation between public decision-makers and the public on compliance and regulation enforcement.

The comments and remarks of the referees were very useful to help us improve the clarity of the text and the coherence between the background section and the methods. We also asked an English native speaker to revise the text.

Below, we will repeat the concerns raised by the three reviewers in the order they appear in the decision letter. Our response to each comment is written in *italics*. Our resubmission includes the file changes.pdf in which we highlight all changes in the main text.

Thank you on behalf of all authors

Dr. Mike Farjam

Referee 1

1.1: I would recommend edits to some of the language throughout to make it clearer or more appropriate. Some examples: Some of the language is not appropriate for a scientific article. For example, describing lock-down and social distancing as “draconian”. I recommend removing this word.

We had an English native language editor checking the text. He is mentioned in the acknowledgments.

1.2: Some sentences I had to read several times to understand. For example, p3, line 49. Recommend re-wording the sentence: “This has led to a consensual understanding of the main factors leading to large-scale cooperation under conditions of strategic interdependence”. Can this be made clearer?

This sentence has been revised. The clarity of text has been improved.

1.3: In methods (a) (line 20), I would change “randomly confronted with one of four vignettes” to “randomly allocated to a group that contains one of four vignettes”

We revised that sentence in combination with comment 1.7

1.4: “trust and legitimacy” I found it hard to read and understand the many sentences that included “trust and legitimacy” added in, when it does not need to be there. For example: see p8, line 51: “Our results confirm that individuals are sensitive to the source of trust and public legitimacy of policy measures”. I’m sure exactly what you are trying to say here, part of the problem is that it’s not clear, but also it may be inaccurate. If you are saying that your results confirm that individuals are sensitive to the source of policy measures, then I agree, and I think this is a much clearer way of describing your results. Adding the words “of trust and public legitimacy of” makes the sentence less clear and potentially misleading as you didn’t measure trust of perceived/actual legitimacy. Rewording the many sentences to describe the “source of information” is much clearer than the “source of trust and legitimacy”. Because ultimately what you are describing is a source credibility effect: the source of information, either politicians, scientists or both.

Good point. Indeed, what we tried to estimate was the effect of the source of information. We have removed any reference to trust and public legitimacy whenever this was misleading.

1.5: P8, line 55. “We found that individuals are more cooperative...”. The authors didn’t measure cooperation, which in this context I would define as working with others to achieve some covid-related goal. It could also mean cooperating with (adhering to) the government guidelines? However, the authors don’t measure either of these things. Please be clearer about which outcomes you are referring to.

Thank you for this comment. This helped us to better define what we meant for cooperation here. We have revised the text in the background section so that any reference to cooperation is clearer throughout the paper now. Note that we also estimated the effect of the manipulation of the source of information on pro-social behaviour, i.e., the willingness of participants to donate to a COVID-related charity instead of having the chance to get a voucher on Amazon.

1.6: Although it is great to see the materials are fully open and accessible, there needs to be further clarification in the paper itself to help the reader understand what you have done. In particular, there should be a section in the methods, subtitle: Measures (or similar), where you can describe the outcomes that you have measured. For example, it appears you have 3

outcomes: donating to covid charity, requesting further info, and support for covid measures. Please add these three in with key details, including: where the measures came from (if you adapted them from existing research), how the variables were measured, how the measures were scored (eg On a scale from 1 to 10 (1 = "completely disagree", 10 = "completely agree")), etc. You can then remove the details in the second paragraph of the results section

We moved that paragraph to a separate "Measurements" subsection in the Method section and added additional detail on our measures.

1.7: Randomisation: In the methods section (a), The authors should add in how the participants were randomised. This is recommended by reporting guidelines for randomised studies eg consort. Presumably in this study the answer is the software that was used to host the questionnaire?

Thank you. We forgot to state that we used oTree, which is a widely-used platform for experimental research guaranteeing correct randomization. The following sentence is now included "The experiment was implemented with oTree, which randomly assigned participants to treatments."

1.8: P9, line 43, you state that covid-19 has probably increased the credibility of scientific experts. Citation is needed for this claim. It's an interesting point, but I don't know of any evidence for this claim. If none exists, then this needs rewording

Thank you. We added a recent reference on the increasing confidence on scientific experts in Italy during the pandemic.

1.9: Questionnaire: It's great to see the questionnaire and other materials in English (for me and fellow English readers). However, given that the original was written in Italian, the Italian version should also be made available, either as an appendix or linked to an online repository.

Based on a comment from referee 3, we uploaded the ethical approval that we got before conducting the experiment. This document includes the original instructions and is made available in our online repository. We explain this when presenting the English instructions in the appendix.

1.10: The authors include interaction effects in the models and report that there are interaction effects on two of the three outcomes. The authors present three graphs in Figure 2 as a way of visualising the interaction term, but they don't appear to interpret these within the results section. All that is said is that there is a "negative interaction" but what does that mean in the current context? More detail of the interactions are needed to fully understand these results. The discussion will likely need updating following these change too.

We have now better explained the interaction effect and its meaning when presenting Figure 2.

Referee 2

2.1: Abstract: Reconsider the use of the phrase 'non-clinical measures'. The WHO have been using the term 'public health measures'

Thank you. We have revised the text.

2.2: In a non-clinical trial setting, should it be 'effectiveness' rather than 'efficacy'

Thank you. We have revised the text.

2.3: Introduction: I would suggest the authors refrain from using the term 'draconian'

We asked a native speaker to check the language and corrected this and some other imprecisions.

2.4: The introduction is written in a way that suggests that the pandemic is over. I would encourage the reframing of the text so that the messages coming across support the notion that they are useful now, not just when things are finished.

Good point. We have revised the text to link our findings to the actual waves of the pandemic. This has been done both in the intro and the closing section.

2.5: Again the sentiment of this sentence is not correct; 'Such norms applied to everyone, including the majority of the population with zero or minimal risk of suffering serious harm from the virus'. It is critical that compliance is high and that public health measures are equitable etc. Beyond this, there is of course the issue of asymptomatic transmission.

Here, we did not make a point about the desired level of compliance. Obviously, we all would like to see max compliance by everyone during a pandemic. However, here we wanted to stress that the cost of anti-COVID regulation compliance was (and still is) unequally distributed across different groups due to different level of vulnerability. Given that some groups could be differently sensitive to compliance costs because less vulnerable against the effect of the virus, it is likely that they could be tempted to free ride on restrictions while benefitting from compliance from others. We clarified the fact that we wanted to focus on possible behavioural responses of the public and cited relevant recent references on these behavioural factors.

2.6: Currently the introduction is very long. Consider moving the section on roles/responsibility (page 4, section 17-40) into the discussion or tighten up the paragraphs. The opening paragraph could also be reduced to a couple of sentences, as the epidemiology/ impact of the pandemic is well documented now.

Thank you for these suggestions. We have tried to shorten the introduction and eliminated an entire redundant paragraph.

Referee 3

3.1: Page 2: Line 21: What is an emergency norm? The term should be explained here I think.

We have revised the text to improve its clarity.

3.2: Line 27: “When strict measures are announced and attempted without having strategies to enforce them, which contemplate efficient communication, public credibility of public decision-makers, and attention to trust formation” Enforce here sounds like a top down approach of control and preventing of ‘rule breaking’. However, the examples seem to point more at internal motivational factors, such as credibility and trust, that relate to compliance. This section should be rewritten to rectify this.

Good point. We have revised the sentence to avoid this misunderstanding.

3.3: Line 29: Language/grammar issue “In order to record public health appliance with norms”. I would suggest having the article proofread by a native speaker.

We asked an English native language editor checking the text.

3.4: Line 39. The authors refer to the covid prevention measures as ‘norms’. I think caution is warranted here. Social norms are an established social psychological construct. A government announcement or creation or enforcement of rules does not necessarily make those rules social ‘norms’. I think the authors should discuss this point more deeply how exactly they justify the use of the term ‘norms’ here or rewrite.

Very good point. We agree. We have specified that we were mentioning regulations not norms. We also revised the background section to focus more on regulation compliance and enforcement so that it is now clear that whenever using cooperation, we were mentioned cooperation on regulation compliance, i.e., paying an individual cost of liberty infringement to benefit the more vulnerable groups and the health system.

3.5: Page 3: Line 7: “promote large-scale cooperation on norm compliance”. Again, I am not understanding clearly why the authors are talking about ‘norm compliance’ here. It needs to be established what the norm is. A solid definition of the terminology in general and specifically what is meant here is needed I think. Otherwise it is not clear what is meant by ‘cooperation on norm compliance’. I am not commenting on any further use of the term ‘norm’, as the authors are carrying this theme through (e.g. talking about norm enforcement etc.). In the rewriting process the whole manuscript should be adjusted accordingly.

See above. We have clarified these points.

3.6: Page 4, Methods: Could the authors elaborate on their hypotheses regarding information seeking behaviour? How is that related to trust in the source? I could see the argument going both ways. If people trust the source more they look for more info, possibly to educate themselves. But they could also be interested if they don’t trust it, just to find out more on it and learn more about potential ‘misinformation’. So I do not see a clear direction of the effect here.

Good point again. It is impossible to understand whether the participants’ decision to seek information on the app was driven from the willingness to check the quality of the content included in the link. However, we believe that what is relevant here is that the source of information influenced the probably of participants to click on the link and so spend time on reading the content (for whatever reason). Considering that participants were more likely to seek information when the source of recommended measures included scientific experts alone, we might assume that they were searching for reliable information. However, again, estimating the motivations behind such decisions was out of the scope of our study.

3.7: It is unclear at what point the information seeking behaviour was measured. The text said 'at the end they were presented with the opportunity to seek more'. The methods section did not specify this further. It ends by saying the lottery was resolved. Was the information seeking measure taken after that or before? If after, I could see there being a bias from people for instance not having won the lottery and thus being in a more negative emotional state which in turn could have influenced their decision to seek further info or not. Emotions, especially positive ones can enhance prosocial behaviour. So I see a risk of bias here. Could the authors elaborate on this please and detail how they avoided such bias and present relevant statistics? Thank you.

The information seeking measure was placed right after the vignette and thus before the lottery was resolved. We now clarified this point in the method section.

3.8: I see a potential confound and bias in their sampling. They used facebook groups and posts on covid 19 and regional issues. It is quite a specific sub population of users who are a) on the internet in times of a pandemic, b) on facebook, c) active on covid and regional groups, d) willing to participate in this research. Yet, the language that the authors use to talk about their results suggests that they are trying to provide insights on the general human relationship to sources of information and trust with regards to politicians and scientists. I am not sure if such broad claims can be made based on the available data. The authors should comment on this and state clearly how they have resolved such biases. It would be ideal if the authors could replicate their study with a different more representative and broad sample. If not, I think it would help to adjust the language and rewrite the claims of the paper.

This is an important point and we agree with the referee that a better random and representative sample would have allowed achieving greater external validity. Nevertheless, as we now better explain in the text, the lockdown measures in Italy in the spring 2020 and our research interest to organise an experiment while the debate was at its peak in the country limited our opportunities to use a more representative online sample (as we now better explain in the text). As you know, a representative population sample has significant cost and longer procedures to be available, which should be justified especially whenever trying to measure the impact of a specific event rather than, for example, a change in opinion dynamics over time. In case of urgent issues, such as the people's response to anti-pandemic measures during the event, research could also value the rapidity and low-cost of data collection online. This is indeed one of the reasons why we adopted an experimental approach with an online study. In these cases, what is lost in terms of sample representativeness could be balanced by manipulations that achieve a consistent level of internal validity. This said, we now revised our conclusions, to better take into account this limitation of the study.

3.9: A power analysis is missing. How do the authors justify their sample? How did they determine that it gives them enough power to detect the effects of interest? I would ask the authors to run a power analysis and detail exactly how they justify estimated effect size and ensuing sample size.

We now include the power analysis on which our sampling was based in the appendix. We show that with the sample we have (270 participants per treatment) we could expect to identify a treatment effect for all dependent variables at a power > 0.8 and a significance threshold of 0.05, given that at least 1-2% of variance is uniquely explained by our treatment manipulation. We considered this the smallest effect size we would be interested in.

3.10: Figure 2: I do not understand the figure fully. The experiment had three groups, politicians only, scientists only and both. I don't understand why the graphs show politicians yes/no on the x axis and scientists yes/no as some kind of interaction type plot. I think it would be clearer if the plot displayed the three groups next to each other.

This is a standard interaction plot, which is a common way to present experimental results. It shows that, the experiment had two independent variables: whether politicians are the source of information (yes/no) and whether scientists are the source of information (yes/no). So our experiment has a full factorial design and thus 4 (not 3) groups. This is why we can identify interaction effects in our regressions and visualize them in figure 2. To make this clear, the method section includes the following sentence: “The source could either be (i) scientific experts, (ii) politicians, (iii) scientific experts and politicians, or (iv) not specified, the latter used as the baseline treatment.”

3.11: Page 6: The authors run quite a few analyses, checking many different variables, e.g. education, political attitude and more, yet no rationale is given for why they are doing it. I see a mild danger of multiple comparison testing and potential for false positives due to the amount of analyses conducted. I would like to see a power analysis that provides back up that the conducted tests are warranted and reliable.

As written in 3.9, we now include such a power analysis in the appendix. We want to emphasize that our experimental approach naturally brings the focus of the analysis on differences between the experimental groups, not on the external characteristics of the participants. In other words, we did not test whether socio-demographic characteristic explain our dependent variable and that formally multiple testing is thus not an issue with respect to these variables. The socio-demographic subgroups are not used to explore further (interaction) hypotheses but to show that the direction of the effect is consistent across subgroups. We write in the result section “Whenever we interpreted estimates in the following paragraphs, we checked whether they were consistent across different demographics. If not mentioned explicitly, we found no qualitative difference across demographic groups.” Our tables therefore always first show overall results and the regressions on the demographic subgroups are only used to explore which subgroups contributed most/least to the overall effect.

3.12: Further on page 6: The authors report an interaction analysis. Here a power justification is even more crucial as necessary sample size to run an interaction is significantly larger compared to main effects investigations, e.g. investigating differences between groups alone. Without a power analysis I am not sure that the reliability and soundness of the results can be adequately determined.

The power analysis, now provided in the appendix, is based on the number of participants PER TREATMENT. Since the identification of an interaction effect requires an extra treatment (i.e., one for the intercept, two for the pure effects of the treatment, one for the interaction effect), we hope to convince the referee that our study is sufficiently powered.

3.13: Did the authors preregister their hypotheses? That would give further back up to their analysis approach of testing many different models. It currently seems a bit arbitrary as to the conducted and reported tests.

We did not formally pre-register our hypotheses. First, we hope to have convinced the referee in 3.11 that the separate regressions on demographic subgroups was done to check for consistency of the effect of the treatments across subgroups and not to test individual hypotheses. Second, we now include a link to the ethical approval that we got before conducting the experiment. Therein, we clearly define our dependent and independent variables and how we operationalize them. In our case, the full factorial design of our experiment with 4 treatments directly translates to a regression with an intercept, two pure effect, and an interaction term. We hope this convinces the referee that we did not arbitrarily pick hypotheses.

3.14: I would like to see effect sizes reported as well. Without effect sizes it is difficult to judge the relevance of the presented findings.

See answer 3.15

3.15: I appreciate the use of Bayesian statistics for data analysis. However, I would like to see additional analysis using frequentist statistics to see whether results match. Bayesian stats heavily depend on set priors, so it would be a good robustness check to perform frequentist statistics to see whether the found effects hold. I see the combination of statistical approaches as a good check of the soundness of reported findings.

The appendix now includes for all Bayesian models additional tables with frequentist statistics. The results of our frequentist and Bayesian estimates are consistent (whenever Bayes-factors are high for an estimate, p-values are low). In the frequentist tables we also provide partial-R² as measures of effect strength. We did not include measures of effect strength in the Bayesian tables since such is generally not recommended in this framework. However, we want to emphasize that since all models only include dummy variables for the treatments, estimates are on the same scale. This generally allows to compare effect strengths also in the Bayesian tables.

3.16: Discussion: In the discussion the authors claim that their result ‘further confirm prospect theory’. I am not quite seeing how this is the case. Prospect theory did not play a major role in the introduction. Furthermore, framing in prospect theory is most prominently about gains and losses, not about sources. If the authors have a different interpretation in mind this should be made more clear here with relevant references.

Good point. We have removed any reference to prospect theory and outlined only the sensitivity of subjects to small variation of information signals.

3.17: The authors suggest that their findings “suggest the importance of political communication as a means to promote trust and public legitimacy of policy measures”. I can’t quite see how this statement follows from the experiment and the data presented. What do the authors mean by political communication? If anything their results show that scientist focused communication might be more beneficial. Again, this point should be made clearer.

Good point. Under recommendations from another reviewer, we have better specified that what we were meaning for trust and public legitimacy was essentially information signals. We have revised also the conclusions to reflect better this point.

3.18: Last sentence: “Our findings suggest that confronted with uncertainty and global crisis, people would appreciate to reduce public ambiguity regarding roles, responsibility and decision-making”. I can’t see how the study results support this claim. The study is not really about uncertainty. There is no uncertainty presented to people; it is about donating, policy support, information seeking. And I don’t see how any of the measures speak to preferences of ambiguity reduction regarding roles and responsibilities. The results show that people might trust communication from scientists more. But the authors did not test how people react to messages that are for instance ambiguous in terms of their sources.

Thank you for this comment. Following a comment from another reviewer, we have better specified that we were concentrated on information signals not on ambiguity of sources. You are fully correct on this. While we agree that one of the main results of our experiment is that scientific experts are trusted more as source of information, we wanted to outline that the negative effect of a mix of scientific experts and politicians is even more intriguing. Why did participants trust scientific experts less as a source of information when accompanied by politicians? We have better specified these points in the concluding section but one of our hypothesis is that such a mix distorted the information signal included in the manipulation.

3.19: Appendix: “Would you like to obtain further information on the sources cited in the above text? [If yes] More information will be available at the end of the questionnaire.” “Would you like to obtain further information on the possible interventions cited in the above question? [If yes] More information will be available at the end of the questionnaire.” This suggests that this measure was collected twice? If so, when and why? And which data was used for the presented analyses and results? The authors should clarify this.

Indeed this was presented in a confusing way. The question regarding additional information was listed twice, because the order of suggested policy interventions was randomized across participants (to avoid order effects). So it was either asked after the first set of suggested interventions or after the second but not after both. We now present this more clearly in the appendix.

3.20: Title: Lastly, I believe the title is not quite appropriate for the manuscript. It seems to somewhat inaccurately represent the findings. I don't see an aspect of 'danger' in their findings. The results show that levels of donating and information seeking vary by source. Is there an element of negative emotional consequences involved? If so, this should be reported. My point is that an absence of e.g. 'good' behaviour does not equal an occurrence of 'negative' behaviour into the other direction.

Thank you. We agree. The title has been revised to reflect more precisely the content of the paper.

3.21: Furthermore, I don't see how the paper is about role ambiguity. It is about different groups, but there is no ambiguity regarding roles involved as roles are clearly stated.

We have revised the text to clarify these concepts and eliminated them whenever misleading.

3.22: Lastly, the measures don't test public policy specifically; they test donation behaviour and information seeking and support of policies. I do not think this can be called effect on 'public policy'.

Thank you. We have revised the title and the text, whenever necessary.

3.23: The title of the submitted manuscript does not match the title in the data repository. This is odd.

Thank you, this has been corrected.

Appendix C

Dear Editor,

Thank you for the acceptance. The only point that we were asked (by reviewer 1) to change in our manuscript is the use of the word “cooperation”. We revised all text where we referred to cooperation - including the abstract - and reformulated it as "pro-social behaviour" and "pro-social attitudes". We include a .pdf and the editable .tex with the highlighted changes.

Best regards,

the authors